# The Effects of Exercise for Cognitive Function in Older Adults: A Systematic Review and Meta-Analysis of Randomized Controlled Trials

**DOI:** 10.3390/ijerph20021088

**Published:** 2023-01-07

**Authors:** Liya Xu, Hongyi Gu, Xiaowan Cai, Yimin Zhang, Xiao Hou, Jingjing Yu, Tingting Sun

**Affiliations:** 1Faculty of Sports and Human Sciences, Beijing Sports University, Beijing 100084, China; 2Key Laboratory of Sports and Physical Health, Ministry of Education, Beijing 100084, China; 3China Institute of Sports and Health, Beijing Sports University, Beijing 100084, China

**Keywords:** cognitive ability, exercise interventions, elder, meta-analysis, RCT

## Abstract

Background: Physical exercise can slow down the decline of the cognitive function of the older adults, yet the review evidence is not conclusive. The purpose of this study was to compare the effects of aerobic and resistance training on cognitive ability. Methods: A computerized literature search was carried out using PubMed, Cochrane Library, Embase SCOPUS, Web of Science, CNKI (China National Knowledge Infrastructure), Wanfang, and VIP database to identify relevant articles from inception through to 1 October 2022. Based on a preliminary search of the database and the references cited, 10,338 records were identified. For the measured values of the research results, the standardized mean difference (SMD) and 95% confidence interval (CI) were used to synthesize the effect size. Results: Finally, 10 studies were included in this meta-analysis. Since the outcome indicators of each literature are different in evaluating the old cognitive ability, a subgroup analysis was performed on the included literature. The study of results suggests that aerobic or resistance training interventions significantly improved cognitive ability in older adults compared with control interventions with the Mini-Mental State Examination (MD 2.76; 95% CI 2.52 to 3.00), the Montreal Cognitive Assessment (MD 2.64; 95% CI 2.33 to 2.94), the Wechsler Adult Intelligence Scale (MD 2.86; 95% CI 2.25 to 3.47), the Wechsler Memory Scale (MD 9.33; 95% CI 7.12 to 11.54), the Wisconsin Card Sorting Test (MD 5.31; 95% CI 1.20 to 9.43), the Trail Making Tests (MD −8.94; 95% CI −9.81 to −8.07), and the Stroop Color and Word Test (MD −5.20; 95% CI −7.89 to −2.51). Conclusion: Physical exercise improved the cognitive function of the older adults in all mental states. To improve cognitive ability, this meta-analysis recommended that patients perform at least moderate-intensity aerobic exercise and resistance exercise on as many days as possible in the week to comply with current exercise guidelines while providing evidence for clinicians.

## 1. Introduction

The ageing of the population is an issue of widespread concern worldwide. The Report on World Population Trends (now referred to as the Report) issued at the 51st session of the United Nations Commission on Population and Development pointed out that the global population would reach 9.8 billion by 2050. The number of older adults over 65 will exceed 1.5 billion, accounting for 16% of the total population. Normal ageing is typically associated with both physical and cognitive decline. Cognitive functioning changes as people grow older. Cognitive function includes memory, language, visual space, execution, calculation, understanding, and judgment [1]. Most of the older adults also experience a cognitive decline to varying degrees, which will not only reduce the quality of life but also affect the basic activities of daily living ability, reduce the remaining life expectancy, and increase the risk of death [2]. Therefore, determining the biological mechanism of cognitive ageing and seeking preventive measures to offset its harmful effects are the current priorities in clinical and public health.

It is hypothesized that the neural and vascular adaptations to physical exercise improve cognitive function through promotion of neurogenesis, angiogenesis, synaptic plasticity, decreased proinflammatory processes, and reduced cellular damage due to oxidative stress. In non-medical therapy, as a low-cost, low-risk, and ready-made intervention, physical exercise has been widely accepted by the public and medical rehabilitation workers [3]. Regular physical exercise was a critical factor in preventing and managing non-communicable diseases. Physical exercise is also conducive to mental health, preventing cognitive decline, depression, and anxiety symptoms, and helps maintain body mass and overall well-being. Many experiments and clinical studies have shown that physical exercise can improve the cognitive function of the older adults [4,5,6]. Regular and active physical activities of the older adults can promote the maintenance, improvement, or rehabilitation of biological processes and slow down the decline of age-related cognitive functions. However, although some experiments have proposed the beneficial effect of physical activity on healthy older adults, there was no specific conclusion at present [7,8]. 

The Mini-Mental State Examination (MMSE) is a global clinical psychological, neuropsychological indicator usually used to screen and evaluate the cognitive status of patients. It can comprehensively, accurately, and quickly reflect the intellectual quality and cognitive impairment of the subjects [9]. However, there are many scales to evaluate cognitive ability. By incorporating different screening methods, we can more comprehensively assess the cognitive function of the older adults. In the past few decades, many scientific research teams have studied the effect of exercise on the cognitive function of the older adults in randomized controlled trials [10,11,12,13,14,15,16,17,18,19,20,21,22,23,24,25,26,27,28,29,30]. However, there were contradictions between the results of these studies, and there was no clear conclusion. This systematic review and meta-analysis intends to explore the following questions: (i) the effects of exercise interventions of aerobic and resistance training modes on cognitive ability in the older adults; and (ii) the effects of exercise on different cognitive task results in the older adults. 

## 2. Materials and Methods

### 2.1. Data Sources and Searches

This article was followed by the Preferred Reporting Items for Systematic Reviews and Meta-Analysis (PRISMA) guideline [31]. A literature search used PubMed, Cochrane Library, Embase SCOPUS, Web of Science, CNKI (China National Knowledge Infrastructure), Wanfang and VIP database to identify relevant articles from inception through to 1 October 2022. This paper searched three classes of keywords: “cognitive”, “older adults”, and “exercise”. The first keywords were “cognition”, “executive function”, “cognitive ability”, “cognitive decline”, and “memory”. The second keywords were “older adults”, “aged”, “old people”, and “elderly people”. The exercise keywords were “exercise”, “physical exercise”, “aerobic exercise”, “strength training”, and “intervention”. The search strategy for PubMed is presented in Table 1. 

### 2.2. Inclusion and Exclusion Criteria 

Literature inclusion was based on evidence-based medicine PICOS framework, mainly considering five factors: participants, intervention measures, control group, research results, and research design [32]. Inclusion criteria were as follows: (i) participants were over the aged of 50 years or older; (ii) the treatment groups were intervention consisting of physical exercise or physical activity; (iii) the control group included routine home care, health education, or lifestyle maintenance; (iv) the outcomes include the use of any standardized neuropsychological instrument to measure cognitive ability, and the statistics include: sample size, mean, and standard deviation; and (v) the studies’ design was strictly limited to randomized controlled trials (RCTs). 

Trials were excluded if they met one of the following exclusion criteria: (i) studies that do not meet the inclusion criteria; (ii) studies without available data for statistics; (iii) conference abstract, observational study, dissertation, or letter; and (iv) exclude articles other than English or Chinese.

### 2.3. Study Selection and Data Extraction

The retrieved literature was screened by three researchers (L.X., H.G., and X.C.) in an independent double-blind way according to the inclusion and exclusion criteria. The first step was to exclude articles that did not meet the inclusion criteria by reading the title and abstract of the literature. The second part was to read and screen the remaining documents in full and determine the final documents to be included. The two researchers (L.X. and H.G.) independently extracted the literature that met the criteria, including the following information: author name; publication year of the article; participants’ characteristics (e.g., age and gender); the number of participants in each group; intervention content; intervention time; intervention frequency; intervention cycle; and reported outcomes. The number of participants, average value, and standard deviation (SD) in each group before and after the intervention training were extracted from the articles included in the analysis.

### 2.4. Quality Assessment

The two authors (L.X. and H.G.) evaluated the methodological quality of the included literature, using the Cochrane Collaboration risk bias assessment tools to assess from the following seven areas [33]: selection bias, performance bias, detection bias, attrition bias, reporting bias, and any other preferences. Each indicator was judged by low bias risk, uncertainty, or high bias risk. Any differences arising from the evaluation process shall be settled by the third arbitrator (X.H.).

### 2.5. Statistical Analysis

Statistical analyses were undertaken using Review Manager 5.4 (Cochrane Collaboration) and Stata version 12.0 (Stata Corp). Since the measurement scores of the exercise group and the control group from baseline to the endpoint are continuous variables, Standardized mean difference (SMD) and 95% confidence intervals (CIs) were calculated according to the mean, standard deviation and sample number of outcomes indicators of the intervention group and the control group. The heterogeneity across the studies was evaluated using the *I*^2^ statistic. *I*^2^ represents the heterogeneity of the study; when *I*^2^ ≤ 25%, it indicated insignificant heterogeneity. The moderate heterogeneity was assessed when *I*^2^ ≤ 50% and *I*^2^ > 25%. When *I*^2^ ≤ 75% and *I*^2^ > 50%, it indicated high heterogeneity [33]. *I*^2^ represents the heterogeneity of the study; if *I*^2^ < 50% or *p* ≥ 0.05, the fixed effect model is used to combine the effects; If *I*^2^ ≥ 50% and *p* < 0.05, the random effect model is used for analysis. 

If there is heterogeneity, a subgroup analysis of regulatory variables is performed. Visual analysis of funnel plot symmetry was used to test publication bias [34]. We conducted subgroup analysis according to the gender and age of the participants, the type of intervention, and the duration of the intervention to further explore the source of heterogeneity. All tests were two-tailed, with inspection level α = 0.05; when the bilateral test *p* < 0.05, it is considered that the difference is statistically significant.

## 3. Results

### 3.1. Search Results

Through searching Chinese and foreign databases, 10,336 articles were preliminarily obtained. Two articles were retrieved manually from other resources. Two independent researchers screened 9185 titles and abstracts after eliminating duplicate published articles. After screening the title and abstract, 9135 research articles were excluded. After filtering titles and abstracts, 9135 articles were excluded. The remaining 50 articles were read in totality, and 21 were included in the final analysis, meeting the requirements of systematic evaluation and meta-analysis. The third reviewer will discuss and decide on any differences in the literature screening process. The search procedure is presented in Figure 1.

### 3.2. Studies Characteristics

A total of 21 RCT research articles were included in the mate-analysis [10,11,12,13,14,15,16,17,18,19,20,21,22,23,24,25,26,27,28,29,30]. All included studies were published between 2000 and 2022, including 1414 participants. The subjects in most studies were mixed-gender groups. The topics in five studies were only women [13,18,20,27,28], and only male participants were included in three studies [22,23,29]. For exercise types, 12 trials performed physical exercise [10,11,12,13,14,15,16,17,18,19,20,21], and 9 tests performed mind–body exercise [22,23,24,25,26,27,28,29,30]. The duration of exercise intervention varies from 8 to 52 weeks, and each study has its own time and frequency of intervention. Details of study characteristics are presented in Table 2.

### 3.3. Quality Evaluation

A summary of the bias risks of all included studies in the meta-analysis is shown in Figure 2A. Figure 2B shows the deviation risk of bias for self-reported and physiological measurement of each included study according to the Cochrane risk of bias tool [33]. These 21 studies have relatively high quality, 12 trials reported the generation process of random sequences, and 5 trials reported the methods used to allocate hiding. Because the subjects have to carry out exercise intervention and cannot be blinded, the performance bias assessment in the study was high risk.

### 3.4. Effects of Exercise on Cognitive Functions

This meta-analysis synthesizes the outcome data of the included studies using the same outcome indicators. Overall, the meta-analysis included the following cognitive abilities outcome indicators: the Mini-Mental State Examination (MMSE), the Montreal Cognitive Assessment (MoCA), the Wechsler Adult Intelligence Scale (WAIS), the Wechsler memory scale (WMS), the Wisconsin card sorting test (WCST), the Trail Making Tests (TMT), and the Stroop Color and Word Test (SCWT). The study found that the exercise intervention methods for the cognitive function of the older adults were mainly divided into two categories: aerobic and resistance exercise. 

Twelve studies reported the outcomes of the MMSE scale. Figure 3 shows that the MMSE score of the exercise group was higher than that of the control group (MD 2.76; 95% CI 2.52 to 3.00; *p* < 0.00001; *I*^2^ = 18%), and the heterogeneity between studies was low. Six articles reported the evaluation outcomes of the MoCA scale. The MoCA scores of the exercise group were higher than that of the control group (MD 2.64; 95% CI 2.33 to 2.9; *p* < 0.00001; *I*^2^ = 0%). There was no heterogeneity between studies (Figure 4). Four studies reported the outcomes of WAIS. Compared with the control group, the WAIS score of the exercise group increased significantly (MD 2.86; 95% CI 2.25 to 3.47; *p* < 0.00001; *I*^2^ = 39%), and the heterogeneity between studies was low (Figure 5). Three studies provided the evaluating outcomes of the WMS scale. Figure 6 shows that the exercise group’s WMS score significantly increased compared with the control group (MD 9.33; 95% CI 7.12 to 11.54; *p* < 0.00001; *I*^2^ = 28%), and the heterogeneity between studies was low. Two studies provided the evaluating outcomes of the WCST scale. Figure 7 shows that the WCST score of the exercise group was higher than that of the control group (MD 5.31; 95% CI 1.20 to 9.43; *p* = 0.01; *I*^2^ = 0%). There was no heterogeneity between studies. Three studies provided the evaluating outcomes of the TMT scale. As shown in Figure 8, the analysis indicates, compared with the control group, a significant decrease in TMT scores in the exercise group (MD −8.94; 95% CI −9.81 to −8.07; *p* < 0.00001; *I*^2^ = 30%). Three studies provided the evaluating outcomes of the SCWT scale. The SCWT scores of the exercise group were lower than that of the control group (MD −5.20; 95% CI −7.89 to −2.51; *p* = 0.0002; *I*^2^ = 0%). There was no heterogeneity between studies (Figure 9).

### 3.5. Publication Bias

A visual inspection of the funnel plots for seven different outcomes (Figure 10) indicated no publication bias for the cognitive ability of the older adults.

## 4. Discussion

This study explored the intervention effect of exercise on the cognitive function of the older adults from the perspective of evidence-based medicine. The results showed that exercise could effectively delay the decline of the cognitive function of the older adults. According to 21 eligible trials, we extracted and analyzed cognitive ability (MMSE, MoCA, WAIS, WMS, WCST, TMT, SCWT scales) outcomes to evaluate the cognitive function change after exercise interventions. The key finding from this study is that physical exercise interventions effectively improve cognitive function in the elder, regardless of mental status. 

The overall analyses suggest evident improvements in exercise on cognitive ability in the elder. The study found that the scores of the cognitive ability assessment scale (MMSE, MoCA, WAIS, WMS, and WCST) were significantly increased in the low heterogeneity and non-heterogeneity intervention groups. Specifically, exercise positively affected cognitive ability by reducing scores on TMT and SCWT scales. 

Studies that included aerobic or resistance training in traditional exercise patterns showed similar results. Some studies [35] have found that changes in carotid artery elasticity and imbalance of vasoconstriction and relaxation function can aggravate the degree of cognitive ability damage. These changes will significantly affect the body’s ability to supply blood and oxygen to brain tissue, causing a large amount of oxygen free radicals to accumulate and damage brain tissue. Therefore, exercise can improve cardiovascular function, increase cerebral blood flow and oxygen supply capacity, give brain tissue cells more nutrition, help maintain brain function, and, thus, delay or reverse the neurodegenerative process and disease tracking. 

Our study suggests that aerobic exercise benefits older adults’ cognitive functioning. Studies have investigated the effects of two short-term exercise intervention plans on various outcome parameters and executive ability of heart rate variability (HRV) in the older adults by Albinet et al. [11]. The results emphasize that aerobic exercise intervention played an essential role in cardio cerebral vascular protection and show a direct relationship between exercise, HRV, and cognition in the older adults. In addition, some studies have also concluded that specific aerobic intervention can improve the cognitive function of the older adults to varying degrees [10,12,13,14,15,16,17,18,19,20,21]. 

This study suggests that resistance training may be essential in improving cognitive function in older adults [36]. Resistance exercise can increase the operation of muscle pumps by squeezing peripheral blood vessels, which can increase the cardiac output per stroke, thus increasing cerebral perfusion. Liu-Ambrose et al. compared the effects of different resistance training models on the cognitive function of the older adults [27]. The results showed that resistance training benefited the older women’s selective attention and the executive effect of cognitive ability. This indicates that the intervention of resistance movement has a particularly significant impact on the cognitive function of these older people [22,23,24,25,26,28,29,30]. 

Previous studies believed that exercise positively impacts the cognitive ability of the older adults, improving memory and inhibition control functions. Exercise is an effective means to treat and intervene in cognitive impairment in the older adults, consistent with the previous review [35,37]. Some studies showed that exercise intervention could improve the cognitive function of the older adults, providing strong evidence for exercise as an effective non-drug intervention. Still, the method of exercise needs to be designed according to the differences between different individuals, which was the main reason for the differences in many research results. In addition, most of the studies were aimed at patients’ physical indicators and cognitive ability, and there were few studies on the mechanism of brain action. 

This paper summarizes the intervention of different intervention methods, different from other studies that outline a single sports event. The utility of different periodic and types of sports was more straightforward, which provided a reference for future research and practical application. This paper summarizes the beneficial evidence of aerobic exercise and resistance exercise in improving the cognitive ability of the older adults points out the positive effect of aerobic exercise and resistance movement on improving mild cognitive impairment and emphasizes that we must follow the scientific and safe principles, adopt reasonable exercise methods, improve the cognitive ability of the older adults through regular scientific exercise, improve the body ability and cardiopulmonary function, and provide conditions for the older adults to maintain continuous training. 

This study was carried out following the PRISMA statement list, but there were still some shortcomings and limitations. First, the search scope for the literature does not include unpublished literature, and some literature is not included due to incomplete outcome index data, which may affect the comprehensiveness of the data to some extent. Meanwhile, the sample size of the meta-analysis fit in the study is small, which may also reduce the reliability of the analysis results. Finally, although two researchers used an independent double-blind method to evaluate the quality of the included literature, they only used the “Cochrane Risk Bias Tool” for evaluation. Due to subjective judgment errors, specific evaluation errors may be caused. Therefore, it is recommended to add other judgment criteria to minimize personal evaluation error.

## 5. Conclusions

This systematic review and meta-analysis demonstrate that regular exercise benefits older adults’ cognitive function. Exercise could be used as a supplementary therapy to treat the cognitive decline of the older adults.

## Figures and Tables

**Figure 1 ijerph-20-01088-f001:**
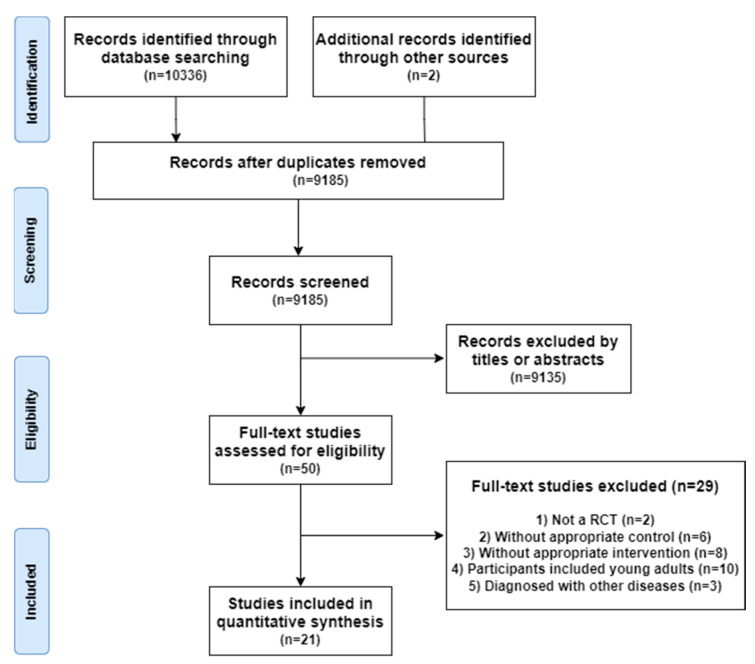
Flowchart representing the selection progress.

**Figure 2 ijerph-20-01088-f002:**
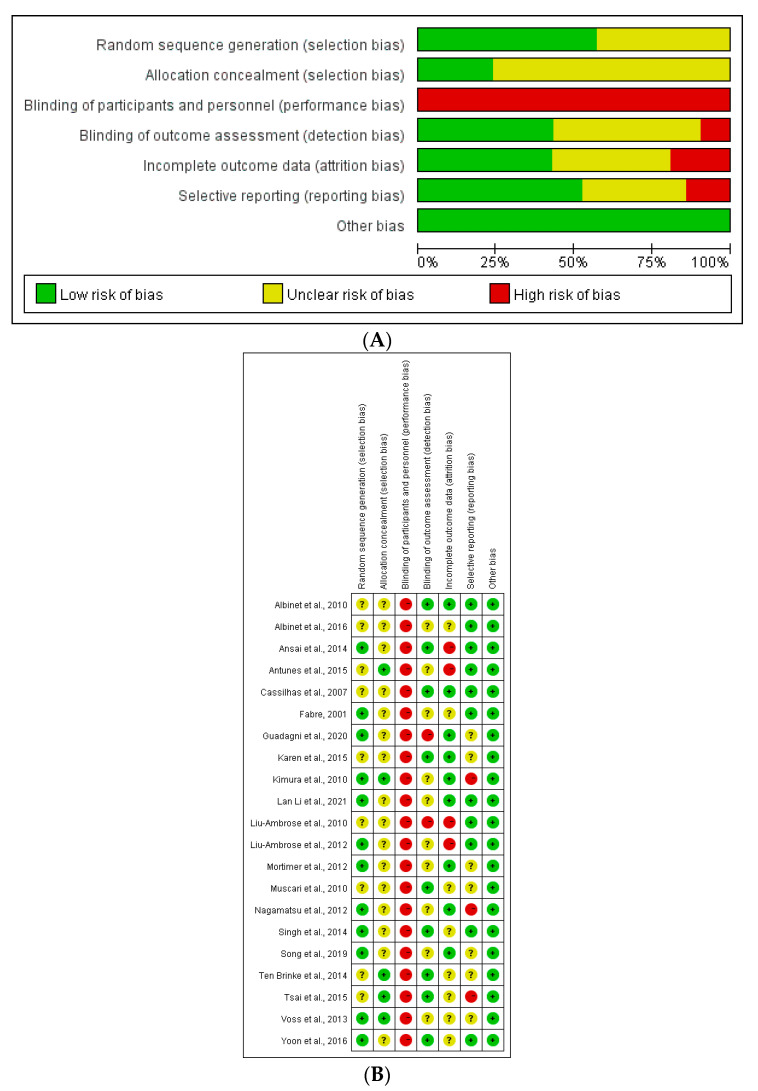
(**A**). Risk of bias summary; (**B**) risk of bias assessments [10,11,12,13,14,15,16,17,18,19,20,21,22,23,24,25,26,27,28,29,30].

**Figure 3 ijerph-20-01088-f003:**
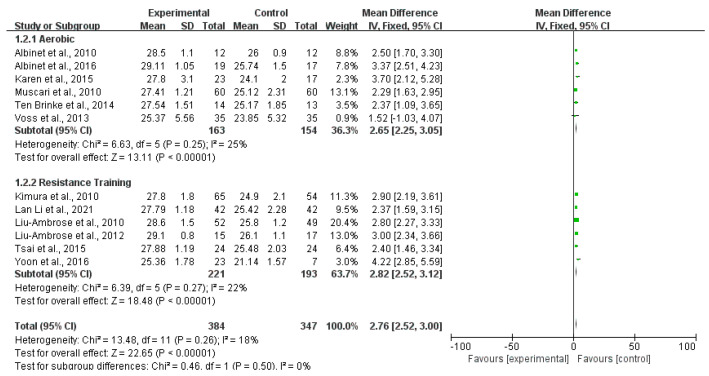
Forest plots of MMSE scale outcomes in overall analysis [10,11,13,17,20,21,25,26,27,28,29,30].

**Figure 4 ijerph-20-01088-f004:**
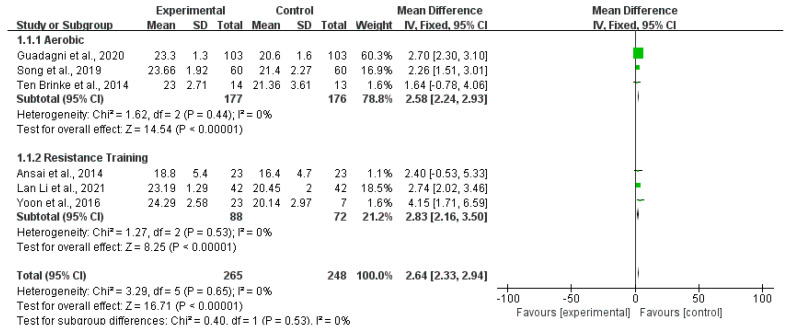
Forest plots of MoCA scale outcomes in overall analysis [15,19,20,22,26,30].

**Figure 5 ijerph-20-01088-f005:**
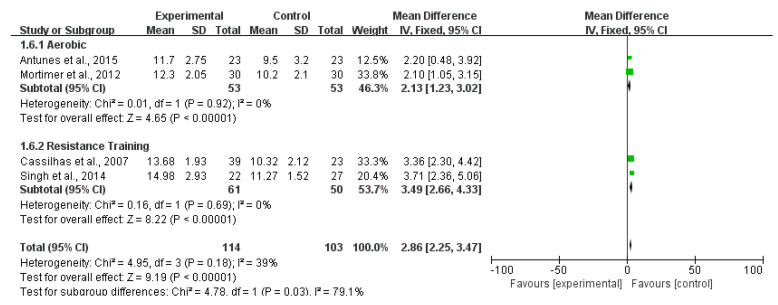
Forest plots of WAIS scale outcomes in overall analysis [12,16,23,24].

**Figure 6 ijerph-20-01088-f006:**
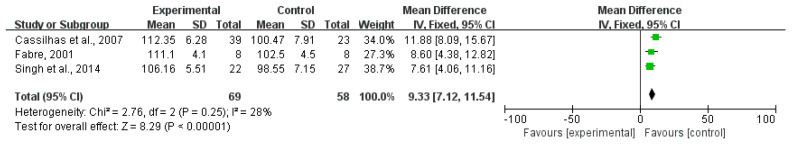
Forest plots of WMS scale outcomes in overall analysis [14,23,24].

**Figure 7 ijerph-20-01088-f007:**
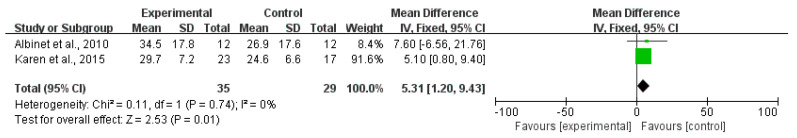
Forest plots of WCST scale outcomes in overall analysis [11,13].

**Figure 8 ijerph-20-01088-f008:**
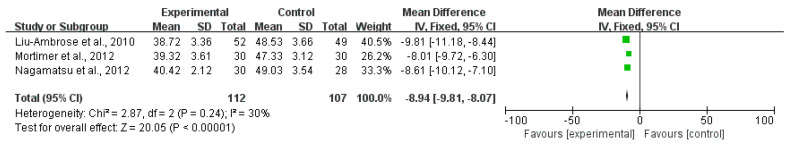
Forest plots of TMT scale outcomes in overall analysis [16,18,27].

**Figure 9 ijerph-20-01088-f009:**
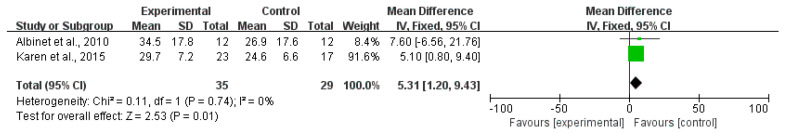
Forest plots of SCWT scale outcomes in overall analysis [11,13].

**Figure 10 ijerph-20-01088-f010:**
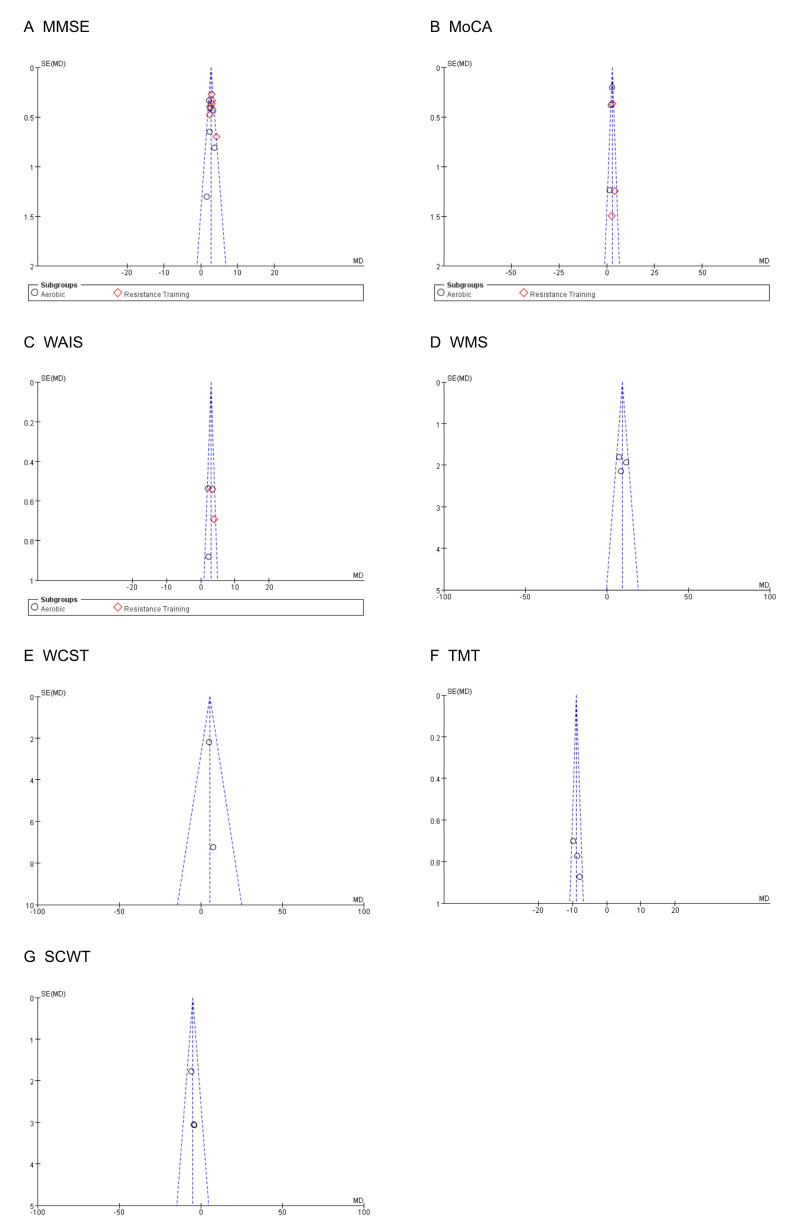
Funnel plot assessing the publication bias.

**Table 1 ijerph-20-01088-t001:** Database search of PubMed.

#	Searches	Results
1	((((“exercise*”[Title/Abstract]) OR (“sport*”[Title/Abstract])) OR (“physical exercise”[Title/Abstract])) OR (“exercise intervention”[Title/Abstract])) OR (“intervention*”[Title/Abstract])	1,586,474
2	(((“older adults*”[Title/Abstract]) OR (“aged*”[Title/Abstract])) OR (“old people”[Title/Abstract])) OR (“elderly people”[Title/Abstract])	930,566
3	(((“cognitive*”[Title/Abstract]) OR (“cognitive ability “[Title/Abstract])) OR (“cognitive function”[Title/Abstract])) OR (“cognitive decline”[Title/Abstract])	443,939
4	1 and 2	127,165
5	3 and 4	10,645
6	Limit 5 to (English language and humans and “all aged (60 and over)”)	2045

**Table 2 ijerph-20-01088-t002:** Characteristics of the included trials and participants.

Included Studies	Mean Age (Years)	Participants (M/F)	Sample Size (N)	Intervention	Intervention Duration (Weeks)	Session Duration	Session Frequency	Outcome Measure
Guadagni et al., 2020 [15]	65.9	206 (101/105)	IG = 103; CG = 103	Aerobic	48	60 min	3 times/week	MoCA
Song et al., 2019 [19]	75.78	120 (30/90)	IG = 60; CG = 60	Aerobic	16	60 min	3 times/week	MoCA
Nagamatsu et al., 2012 [18]	75.36	58 (0/58)	IG = 30; CG = 28	Aerobic	26	60 min	2 times/week	TMT
Ten Brinke et al., 2015 [20]	75.78	27 (0/27)	IG = 14; CG = 13	Aerobic	26	60 min	2 times/week	MMSE; MoCA
Voss et al., 2013 [21]	64.87	70 (25/45)	IG = 35; CG = 35	Aerobic	52	40 min	3 times/week	MMSE
Albinet et al., 2010 [11]	70.65	24 (11/13)	IG = 12; CG = 12	Aerobic	12	60 min	3 times/week	MMSE; WCST
Fabre, 2002 [14]	65.55	16 (3/13)	IG = 8; CG = 8	Aerobic	8	60 min	2 times/week	WMS
Mortimer et al., 2012 [16]	68	60 (20/40)	IG = 30; CG = 30	Aerobic	40	50min	3 times/week	TMT; WAIS; SCWT
Muscari et al., 2010 [17]	69.2	120 (62/58)	IG = 60; CG = 60	Aerobic	52	60 min	3 times/week	MMSE
Albinet et al., 2016 [10]	66.53	36 (10/26)	IG = 19; CG = 17	Aerobic	21	60 min	2 times/week	MMSE; SCWT
Antunes et al., 2015 [12]	66.97	46 (46/0)	IG = 23; CG = 23	Aerobic	26	60 min	3 times/week	WAIS
Karen et al., 2015 [13]	64.58	40 (0/40)	IG = 23; CG = 17	Aerobic	26	60 min	3 times/week	MMSE; WCST
Lan Li et al., 2021 [26]	70.48	84 (33/51)	IG = 42; CG = 42	Resistance	24	30min	5 times/week	MMSE; MoCA
Tsai et al., 2015 [29]	71.4	48 (48/0)	IG = 24; CG = 24	Resistance	52	60 min	3 times/week	MMSE
Yoon et al., 2017 [30]	76	30 (not stated)	IG = 23; CG = 7	Resistance	12	60 min	2 times/week	MMSE; MoCA
Liu-Ambrose et al., 2010 [27]	69.62	101 (0/101)	IG = 52; CG = 49	Resistance	52	60 min	2 times/week	MMSE; TMT; SCWT
Liu-Ambrose et al., 2012 [28]	69.31	52 (0/52)	IG = 15; CG = 17	Resistance	52	60 min	2 times/week	MMSE
Cassilhas et al., 2007 [23]	68.08	62 (62/0)	IG = 39; CG = 23	Resistance	24	60 min	3 times/week	WMS; WAIS
Kimura et al., 2010 [25]	74.33	119 (49/70)	IG = 65; CG = 54	Resistance	12	90 min	2 times/week	MMSE
Ansai et al., 2015 [22]	82.7	46 (16/30)	IG = 23; CG = 23	Resistance	16	60 min	3 times/week	MoCA
Singh et al., 2014 [24]	70.1	49 (33/16)	IG = 22; CG = 27	Resistance	26	75min	2–3 times/week	WMS; WAIS

M, man; W, woman; IG, intervention group; CG, control group; MMSE, the Mini-Mental State Examination; MoCA, the Montreal Cognitive Assessment; WAIS, the Wechsler Adult Intelligence Scale; WMS, the Wechsler Memory Scale; WCST, the Wisconsin Card Sorting Test; TMT, the Trail Making Tests; SCWT, the Stroop Color and Word Test.

## Data Availability

The data presented in this study are openly available in the studies referenced in the figures. The individual data in each can be seen in the original manuscripts.

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
