# Peer review of "The Effects of Exercise for Cognitive Function in Older Adults: A Systematic Review and Meta-Analysis of Randomized Controlled Trials"

_ijerph, 2023, doi:10.3390/ijerph20021088_

Round 1
Reviewer 1 Report (Previous Reviewer 1)
The paper presents a review with meta-analysis of randomized controlled trials on the effects of exercise for cognitive function in the older adults.
The manuscript is well written and the topic of interest.
Several issues could be improved:
The conceptual rationale could be elaborated in more detail. For instance, which physiological / neural mechanisms may be most important? How were the Preferred Reporting Items for Systematic Reviews and Meta-Analysis (PRISMA) guidelines been addressed?
Only ten studies were included. This number is extremely small and almost not enough to run a proper meta-analysis with multiple subgroups. Results may have to remain explorative, especially given that this field of research is quickly advancing and more literature will surely be available in the near future.
The authors wrote “The purpose of this study was to compare the effects of aerobic and resistance training on cognitive ability.” Maybe I misunderstood, but it seems that they did not compare those two training types with each other. Instead, the two training types were compared with a control condition. This needs to be made clearer throughout the
manuscript, including the abstract.
The quality of the included studies could be discussed in more detail.
Were all studies of sufficient quality?
What are the specific recommendations for future research? What needs to
be addressed next? How research and understanding of the underlying
mechanisms can be advanced?
Author Response
Please see the attachment.

Reviewer 2 Report (Previous Reviewer 2)
· The Revised paper reports a thorough work and is very clear. The improvements in the text are well done.
· The paper has a small typo. Below I also suggest a small improvement in update of the literature.
1. Lines 7, 8, 12, 13, 20, 29, 30, ... and further in the text: for readability you should put a blank for every opening bracket. Stead of ‘ com(L.X.)’ you should have ‘ com (L.X.)’. Compare your lines 9, 10, 13, 23, 28, … Please make the text consistent. This remark only applies to the beginning of the text, say the first 100 lines and does not hold for the references.
2. Lines 58,59, I think your references are not up-to-date here. I recommend to also include [1,2] below, in your references.
References update:
1. Pedroli, E.; Greci, L.; Colombo, D.; Serino, S.; Cipresso, P.; Arlati, S.; Mondellini, M.; Boilini, L.; Giussani, V.; Goulene, K.; et al. Characteristics, Usability, and Users Experience of a System Combining Cognitive and Physical Therapy in a Virtual Environment: Positive Bike. Sensors 2018, 18, 1–16.
2. Koppelaar, H.; Kordestani-Moghadam, P.; Kouhkani, S.; Irandoust, F.; Segers, G.; Haas, L. de; Bantje, T.; Warmerdam, M. van Proof of Concept of Novel Visuo-Spatial-Motor Fall Prevention Training for Old People. Geriatrics 2021, 6, 1–27.
Round 2
Reviewer 1 Report (Previous Reviewer 1)
The manuscript entitled "The Effects of Exercise for Cognitive Function in the older adults: A Systematic Review and Meta-Analysis of Randomized Controlled Trials" has a significant contribution, is well organized, and comprehensively described. The comments were partly answered. I do not have further comments about the papers. Congratulations o the authors.
This manuscript is a resubmission of an earlier submission. The following is a list of the peer review reports and author responses from that submission.
Round 1
Reviewer 1 Report
The paper presents a systematic review with meta-analysis of randomized
controlled trials investigating potential effects of exercise on
cognitive functions in older adults.
The paper is well written and the results of interest.
However, there are several major issues:
“Elderly” and “elders” are inappropriate wordings (stereotypes /
prejudices) – please use more neutral wordings like “older adults”.
“cognitive function” is an inappropriate wording, because it does not
exist one cognitive function to describe the manifold cognitive
abilities in humans. Better would be to specify the different cognitive
measures / abilities targeted.
From 1000 matches only 10 studies were included. Are those
representative of this huge field of research?
How does this study deal with the massive heterogeneity across the
studies? Is a meta-analytical approach really applicable since it
requires a minimum of homogeneity across studies.
The search was done on June 1, 2022 – probably more studies have been
published since then in this rapidly growing field.
The main conclusion that “physical exercise improved the cognitive
function of the elderly in all mental states. To improve cognitive
function, this meta-analysis recommended that patients perform at least
moderate-intensity aerobic exercise and resistance exercise on as many
days as possible in the week to comply with current exercise guidelines”
is not super novel and just reflects what has been discussed since
decades. The novelty of the study needs to be made substantially clearer.
Reviewer 2 Report
Conclusions
· The paper reports thorough work and is very clear.
· The language used in the paper is good, but has here and there a small typo. Below I suggest a few improvements.
Observations and critical remarks
1. Title: ‘Elders’ should read: ‘Elderly’.
2. Title: it is too long for easy reference. I suggest making it shorter, to enhance your future exposure to the literature. For instance: Review of Exercise Effects for Cognitive Functioning of Elderly.
3. In Lines 6 – 13, your shortcuts for author names are punctuated, while in the paper, lines 105, 110, 118, 123, and at the end of your paper in lines 309 – 312 they are not. Please remove the dots, to make it consistent.
4. Line 47 and further references in the text: for readability you should put a blank for every reference opening bracket. Stead of ‘judgement[1]’ you should have ‘judgement [1]’. And everywhere further.
5. Line 82, ‘This paper searched three keywords’ is wrong. You could say here ‘This paper searched three classes of keywords’.
6. Lines 133, 141, 196, 202, and further, need a blank before and after the mathematical operators <, =, > . The text is now inconsistent at this point, as you can see in line 133, where ‘12≤ 50%’ should read ’12 ≤ 50%’. (Compare with your correct blanks in Lines 134,135,142.) Also in line 133 ‘P>50%’ should read ‘P > 50%’. And another example: in Line 141 should a blank be typed before the = sign.
7. Line 137, ‘the’ should be deleted in ‘the heterogeneity’.
8. Line 249, please insert ‘scores on’ in ‘cognitive function by reducing TMT and SCWT scales’: ‘cognitive function by reducing scores on TMT and SCWT scales’.
9. Line 282, ‘differences of’ should read: ‘differences between.
10. Lines 303 - 304, I recommend ameliorating your final sentence, by saying that ‘Therefore, it is recommended to add other judgment criteria to minimize personal evaluation error.’
11. Lines 320 – 400, in the Literature: you have the shortcut ‘[J]’ in references. It seems meaningless to me. Please remove or explain its purpose. The use of capitals for the author names does not obey the style of the journal. Also, other adaptations are necessary for the references (bold years, etc.)
